# A Diet Supplemented with Oil-Free Olive Pulp Flour (MOP-ManniOlivePowder^®^) Improves the Oxidative Status of Dogs

**DOI:** 10.3390/ani14172568

**Published:** 2024-09-04

**Authors:** Sara Minieri, Iolanda Altomonte, Virginia Bellini, Lucia Casini, Angelo Gazzano

**Affiliations:** Dipartimento di Scienze Veterinarie, Università di Pisa, 56124 Pisa, Italy; sara.minieri@unipi.it (S.M.); virgy.bellini95@gmail.com (V.B.); lucia.casini@unipi.it (L.C.); angelo.gazzano@unipi.it (A.G.)

**Keywords:** circular economies, dog feeding, olive flour, oxidative stress, d-ROMS, OXY-adsorbent test

## Abstract

**Simple Summary:**

Olive oil production generates high amounts of coproducts that can be exploited for other purposes. Olive oil coproducts are rich in polyphenols, which have beneficial effects when added to cow, pig, and chicken diets. Products rich in polyphenols also have potential benefits for animal welfare through many mechanisms. These include antioxidant activities and a reduction in oxidative stress, which is linked to aging and many diseases. To the best of our knowledge, there are no studies on the use of olive oil flour in dog diets. First, we demonstrated that introducing a dose of 11.5 mg/kg body weight of olive flour rich in polyphenols (MOP^®^) to the diet of dogs decreased oxidative stress, reduced the levels of d-ROMs, and led to increasing trends in the amount of blood antioxidants.

**Abstract:**

Olive oil coproducts and their phenolic extracts have shown beneficial effects when added to the diets of food-producing animals, whereas data on their effects on pets are scarce. The aim of this study was to evaluate the effects of dietary supplementation with olive flour (MOP^®^) on oxidative blood biomarkers in dogs. Thirty dogs were recruited and divided into two groups. Both groups were fed the same kibble feed twice daily. The treatment group (T) also received canned wet feed supplemented with 11.5 mg/kg of body weight of organic olive flour per day, whereas the control group (C) received the same wet feed without any supplementation. The findings showed that oil-free olive pulp flour supplementation led to a significant decrease in d-ROMs (*p* < 0.044) in the blood of the T group (from 101.26 to 86.67 U CARR), whereas no significant changes were observed in the C group. An increasing OXY trend was found in the blood of the T group. Polyphenols in olive flour at a dose of 11.5 mg/kg of body weight contributed to lowering the oxidative stress threshold in dogs, reducing the levels of d-ROMs in dogs and leading to increasing trends in the amount of blood antioxidants. The use of olive pulp flour in dog diets has proven to be beneficial for their health and could also reduce the waste associated with olive oil production.

## 1. Introduction

Olive oil production generates high amounts of coproducts that should be exploited. These coproducts contribute to the circular economy of Mediterranean countries, which are the main olive oil producers and may have environmental and economic benefits. Olive oil coproducts contain several beneficial compounds, and their phytochemical content varies according to the type of product [1]. Polyphenols are among the different bioactive molecules in olive oil coproducts. Polyphenols encompass more than 8000 molecules. They contain benzene rings, with one or more hydroxyl substituents, and range from simple phenolic molecules to highly polymerized compounds.

Each group of molecules has different mechanisms of action correlated to a structural specificity, which confer the antioxidant properties to the compounds. They act mainly in scavenging reactive species of oxygen, nitrogen, and chlorine or can also chelate metal ions, acting in both the initiation stage and in the propagation of the oxidative process. [2,3].

Olive oil coproducts and their phenolic extracts have shown many beneficial effects when added to the diets of food-producing animals [4], such as monogastric animals (pigs, chickens, and rabbits) [1,5] and ruminants [6,7]. In vitro and in vivo studies have also shown that polyphenols improve recovery from pathologies [8] and extend the lifespan in various model organisms [9], with potential beneficial effects on welfare and longevity. The many mechanisms associated with the beneficial effects of polyphenols from olive coproducts include the modulation of gene expression, immune functions, antimicrobial actions, and antioxidant activities [10,11].

Enzymes and molecules, some of which are derived from the diet (e.g., vitamins and carotenoids), have been identified as antioxidants in the body. On the other hand, oxidants are normally metabolic products, such as reactive oxygen species (ROS), and are formed during the enzymatic reactions of intercellular and intracellular signaling. However, some physiological and/or pathological mechanisms can lead to an overproduction of ROS [12].

The balance between oxidants and antioxidant defenses determines the degree of oxidative stress, which is related to cellular senescence and death and is implicated in the pathogenesis of many diseases and inflammatory conditions [12]. For instance, acute and chronic gastrointestinal diseases [13], reproductive issues [14], cognitive dysfunction syndrome [15], and neoplasia in dogs [16] are characterized by altered redox homeostasis due to either an overproduction of reactive oxygen species (ROS) or a deficiency in counteracting antioxidant systems. The role of oxidative stress in the pathogenesis of canine leishmaniosis has been suggested [17]. Pugliese et al. [18] hypothesized that treating oxidative stress in dogs with antioxidant supplementation may improve the clinical outcomes of some diseases.

Dogs exhibit age-related increases in oxidative stress under many chronic conditions, many of which exhibit dynamics similar to those of humans [19]. It has been suggested that dogs could be models of human aging. Trials on the use of polyphenols in pet animals are scarce [13]. To the best of our knowledge, there are no studies on the use of olive oil flour in the diet of dogs. The aim of this study was to evaluate the effects of dietary supplementation with organic olive flour (MOP^®^), which is rich in polyphenols, on oxidative blood biomarkers in dogs.

## 2. Materials and Methods

### 2.1. Ethical Statement

The animal study protocol was approved for nonexperimental zootechnical practice according to D. lgs. vo 26/2014 and was approved by the Ethics and Animal Welfare Committee of The Department of Veterinary Sciences, University of Pisa (resolution n 4/2023 of 24 January 2023).

### 2.2. Animal Recruiting and Diet Composition

The nutritional trial was carried out at two shelter kennels located in the districts of Pisa and Lucca (central Italy), both of which maintained identical health, feeding, and space management. Thirty dogs were recruited following clinical evaluation, similar in weight (between 15 and 30 kg), and aged between 2 and 8 years.

Each dog was kept in an individual 12 m^2^ enclosure (3 × 4) comprising an enclosed resting area of 2 m^2^ (1 × 2) and an open area. All animals were confirmed to be in good health and had undergone haemato-biochemical tests with positive results.

The recruited dogs were divided into two groups, each consisting of 15 animals: control (C) and treatment (T). The dogs of both C and T groups were fed the same dry commercial pet food and wet food twice daily according to individual maintenance energy requirements, with the only difference being the supplementation of organic olive flour (MOP-ManniOlivePowder^®^) rich in polyphenols. To encourage the intake of the olive powder, the T group also received a small amount of canned pet food once a day, supplemented with 11.5 mg/kg of body weight (BW) of organic olive flour, which is the same dose of polyphenols that has been reported as having antioxidant activity in humans. [20]

The C group received the same canned food without any supplementation. The experimental period lasted 3 months (15 days of transition to the new diet and 75 days of the trial).

### 2.3. Chemical Analyses

The commercial food was according to the AOAC (Association of Official Analytical Chemists) International standard methods (method 950.46 for water, 954.01 for crude protein, 920.39 for ether extract, 942.05 for ash and 962.09 for crude fiber; AOAC). The nitrogen-free extract (NFE) was calculated by the difference between the amount of dry matter and the total amounts of crude protein, crude fat, and ash. The metabolizable energy (ME) was evaluated by the modified Atwater method.

MOP^®^ contained 14,856 mg/kg total polyphenols, which was previously reported by Cecchi et al. [21]. MOP^®^ consists of a micronized olive mill byproduct derived from cold pressing of the olives to obtain extra virgin olive oil, resulting in zero waste in the production process.

The dietary analytical constituents are reported in Table 1.

### 2.4. Blood Sampling and Analysis

A cruelty-free method was used for blood sample collection. At the beginning and end of the experimental tests, as part of the routine health checks within the kennel, blood samples were collected for analyses of oxidative stress levels, including reactive oxygen metabolite-derived compounds (d-ROMS) and the OXY-adsorbent test.

### 2.5. Measurements of Oxidative Stress

d-ROMS were assayed with a SLIM (simple, low-power, inexpensive, microcontroller-based) spectrophotometer (SEAC, Calenzano, FI, Italy) using reagents purchased from Diacron (Diacron International SRL, Grosseto, Italy) as described by Pasquini et al. [22]. In the d-ROM test, reactive oxygen metabolites in the biological sample, with iron released from plasma proteins by an acidic buffer, generate alkoxyl and peroxyl radicals according to the Fenton reaction. These radicals can then oxidize an alkyl-substituted aromatic amine (N, N-diethyl-paraphenylenediamine), which produces a pink-colored derivative that is photometrically quantified at 505 nm. The d-ROM concentration is directly proportional to the color intensity and expressed as Carratelli units (1 CARR U = 0.08 mg hydrogen peroxide/dL).

### 2.6. OXY-Adsorbent Test

The OXY-adsorbent test determines antioxidant capacity or potential by quantifying the body’s antioxidant barriers and measuring the ability of a plasma sample to resist massive oxidant attack induced in vitro.

The OXY-adsorbent test (Diacron International, Grosseto, Italy) quantifies the ability of plasma nonenzymatic antioxidant compounds to counter the in vitro oxidant action of hypochlorous acid (HOCl), which is an oxidant produced endogenously. This assay can quantify the contribution of several types of antioxidants, as HOCl reacts with proteins, thiols, ascorbate, vitamin E, and carotenoids. The procedure was conducted as in previous studies [23].

The oxidative stress index (OSI) is considered an index of the plasma redox status and was calculated as the ratio between the values of d-ROMs and OXY multiplied by 100.

### 2.7. Statistical Analysis

Shapiro–Wilk test for normality and Levene’s test for homogeneity of variance were performed on each variable. Data were analyzed via ANOVA, using JMP software version 5 [24], considering treatment, time, and interaction between treatment, and time as fixed effects. The significance level was set at *p* < 0.05.

## 3. Results and Discussion

Throughout the entire experimental trial, Group B dogs showed no adverse effects from the olive powder (e.g., loss of appetite, diarrhea, vomiting, or any other potential issues). Before supplementation with MOP^®^, the d-ROM levels of both groups (Table 2) were similar but slightly higher than the normal range (from 50 to 90 CARR U) reported for dogs [22], corresponding to a condition of borderline/mild oxidative stress [25].

D-ROM values have been shown to change under pathological conditions [26] and with physical exercise [27] and depend on the animal species [28]. In particular, d-ROM levels in dogs are on average less than one-third of those detected in humans (250 to 300 CARR U) [29].

The borderline/mild oxidative stress we observed was similar to that found in previous papers [30] in shelter dogs living in conditions with sufficient space and high-quality feeding. Passantino et al. [30] also suggested that dROM could be a tool for investigating the response of sheltered dogs to different social and spatial restrictions and management practices and could serve as a prognostic tool for assessing welfare and health in sheltered dogs.

The borderline/mild oxidative stress conditions found in this study could suggest that, although maintained in good conditions, shelter dogs tend to experience greater oxidative stress than family dogs. This could be linked to increased physical activity and lack of rest. Dogs in kennels show a progressive significant increase in the duration of activity-related behavioral patterns [31].

Although the effect of treatment was not significant, a significant interaction was found between treatment and time, suggesting that prolonged supplementation of olive pulp flour could have led to a significant decrease in d-ROMs (*p* < 0.044) in the treated group, with a reduction of 16 ± 4%.

On the other hand, no significant changes were observed in the control group. 

Although the outcomes of this study should be considered alongside the limitations, because of the small sample size and the relatively short period of supplementation, the decrease in d-ROMs observed in our study is consistent with findings by other authors [25] who have shown that plant-enriched diets (including *Punica granatum*, *Valeriana officinalis*, *Rosmarinus officinalis*, *Tilia* spp., *Crataegus oxyacantha* L. Tea extracts), which are high in antioxidants, induce a significant decrease in plasma levels of dROMs in elderly dogs. The authors also suggested that diet might be a valuable strategy to counteract aging-related decline in dogs [25].

The OXY-adsorbent test allows for the measurement of the chemically active antioxidant capacity (antioxidant barriers/scavengers) exerted by plasmatic structural components such as mucopolysaccharides, amino acids, and proteins [32,33].

Investigations on rats have reported that daily intake of antioxidant supplements, particularly phenolic compounds, increases circulating antioxidants and could serve as a preventive measure against the risk of stroke onset and symptom severity [34].

In the present study, the OXY values were similar to the average values (HClO/L mmol/L) reported for dogs [35,36]. The OXY-adsorbent test revealed that serum from treated dogs presented higher values, although the difference was not significant (Table 2). This trend between pre- and post-treatment is indicative of a possible increase in the amount of blood antioxidants.

Oxidative stress is an imbalance between oxidants and antioxidants and can be represented by the oxidative stress index (OSI). OSI offers a comprehensive evaluation of the degree of oxidative stress, as high values of OSI indicate a discrepancy between oxidant and antioxidant structures [18].

In our study, we observed no significant changes in the OSI, although there was a tendency for the OSI to decrease in the treated groups.

## 4. Conclusions

This study demonstrates, for the first time, the effects of olive pulp flour on dog health. Olive pulp flour (MOP-ManniOlivePowder^®^), a byproduct of olive oil production that is rich in nutrients and antioxidants, helps lower the oxidative stress threshold in dogs. Olive pulp flour was shown to reduce the levels of reactive oxygen metabolite-derived compounds from initial borderline oxidative stress conditions to normal range values.

The values of the d-ROMs test are a reliable reflection of the state of activity of endogenous oxidative (cellular respiration) and reactive (inflammation) processes and, therefore, of the rate at which the physiological aging process is proceeding at that specific moment. Antioxidant supplements in the diet play a role in maintaining oxidative balance in dogs and could help protect blood lipids from oxidative stress, thus counteracting the oxidative action of free radicals and the cellular damage they generate and therefore potentially protecting animal health. However, the duration of administration can affect the efficacy of supplementation.

The use of olive pulp flour in dog diets has proven to be not only beneficial for their health but also important from the perspective of the circular economy. The integration of olive pulp flour could contribute to reducing the waste associated with olive oil production (currently estimated overall at 20 Mt/year of dry biomass), which can cause significant environmental problems.

The adoption of olive flour in animal diets could stimulate the production of new foods with nutraceutical effects, expanding the market for pet products and offering new economic opportunities.

Future research could develop insight into the use of olive pulp flour in parallel with additional lifestyle improvements, as well as in preventative interventions in dog aging.

## Figures and Tables

**Table 1 animals-14-02568-t001:** Diet analytical constituents (as-fed basis).

	Dry Commercial Pet Food	Wet Commercial Pet Food
Moisture %	8	81
Crude protein %	26	6.0
Ether extract%	16	4.5
Crude fiber %	3	0.5
Crude ash%	6.9	2.6
Nitrogen-free extract	40.1	5.4
ME * kcal/kg	3673.5	781.5

* Metabolizable energy (ME) evaluated by modified Atwater method.

**Table 2 animals-14-02568-t002:** Values of d-ROMs and OXY levels in the two different groups (control and treated) over time.

	Control Group T = 0	Treatment Group T = 0	Control Group T = 1	Treatment Group T = 1	Effect of the Treatment	Time	Time X Treatment
d-ROMs ^1^ (U CARR)	111.83 ^A^	120.04 ^A^	104.14 ^AB^	93.59 ^B^	0.9196	0.0061	0.0444
OXY ^2^ (µmol HClO/L)	126.07	129.36	135.26	187.25	0.3199	0.2780	0.3377
OSI ^3^ %	96.95	100.18	92.72	74.44	0.2633	0.4354	0.2841

^1^ d-ROMs: reactive oxygen metabolite-derived compounds. ^2^ OXY-adsorbent test. ^3^ OSI: oxidative status index. ^A,B^: Within a row, means without a common superscript differ at *p* < 0.05.

## Data Availability

The datasets generated during and/or analyzed during the current study are available from the corresponding author upon reasonable request.

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
