# Peer review of "A Diet Supplemented with Oil-Free Olive Pulp Flour (MOP-ManniOlivePowder®) Improves the Oxidative Status of Dogs"

_animals, 2024, doi:10.3390/ani14172568_

Round 1

Reviewer 1 Report

Comments and Suggestions for Authors

The paper titled "A diet supplemented with oil-free olive pulp flour (ManniOlivePowder®) improves the oxidative status of dogs" investigates the impact of dietary supplementation with olive flour on oxidative blood biomarkers in dogs. The study finds that supplementing dog diets with olive flour rich in polyphenols can significantly decrease oxidative stress, as measured by d-ROMs, and shows an increasing trend in blood antioxidants. This research contributes to understanding the potential benefits of using olive oil by-products in pet nutrition and highlights the positive implications for animal health and environmental sustainability.

Line 43: The term "phenolic rings" should be further explained, as it may not be immediately clear to all readers what these compounds are and their relevance to oxidative stress.

Line 85: The inclusion criteria for the dogs (weight and age range) should be justified in the context of the study. It would be helpful to explain why these specific parameters were chosen.

Line 102: To enhance clarity and provide context for all readers, it would be beneficial to include the full meaning of the acronym AOAC in the manuscript.

Line 145: Specify the exact statistical methods used in the ANOVA analysis. Clarifying whether assumptions were checked (e.g., normality, homoscedasticity) would enhance the scientific rigor.

Line 155 (Table 2): it would be helpful to include a footer that provides the full meanings of all acronyms used. This addition would enhance the clarity of the table and ensure that all readers, regardless of their familiarity with the specific terminology, can fully understand the data presented.

Line 178: The reported 16% reduction in d-ROMs should include confidence intervals to provide a sense of the variability and precision of this estimate.

Abstract

The abstract effectively summarizes the study's aim, methodology, key findings, and implications. It concisely presents the novel aspect of the research, using olive pulp flour in dog diets, and highlights the significant reduction in oxidative stress markers. However, it could benefit from more detailed information on the study's methodology and the broader impact of the findings on pet nutrition and sustainability.

Introduction

The introduction provides a clear background on the importance of utilizing olive oil by-products and their potential health benefits. It effectively sets the context for the study and highlights the novelty of applying these by-products to dog diets.

Materials and methods

The statistical analyses used in the manuscript are generally appropriate for the study design and objectives. However, to enhance the rigor and transparency of the research, it would be advisable to include more details on the assumptions of the statistical tests and supplementary metrics such as confidence intervals. These additions would provide a more comprehensive understanding of the data and its implications.

Results

The results section clearly presents the findings, with tables providing detailed data on oxidative stress markers. Providing confidence intervals or effect sizes alongside p-values would strengthen the interpretation of the results.

Discussion

The discussion appropriately interprets the results in the context of existing literature, highlighting the significance of the findings and their implications for pet nutrition and sustainability. The authors make relevant connections to the potential benefits of olive flour for reducing oxidative stress in dogs. However, the discussion could be improved by addressing the limitations of the study, such as the sample size and the generalizability of the findings, and by suggesting directions for future research.

Conclusion

While the conclusion mentions the potential for economic and environmental benefits, the specifics of how olive pulp flour contributes to these areas could be expanded. For example, quantifying potential waste reduction or cost savings would provide concrete evidence of these benefits.

Author Response

The authors would like to thank the Editor and the Reviewers for their time and comments that helped improve the manuscript.

Answers to the Reviewer 1

Comments and Suggestions for Authors

The paper titled "A diet supplemented with oil-free olive pulp flour (ManniOlivePowder®) improves the oxidative status of dogs" investigates the impact of dietary supplementation with olive flour on oxidative blood biomarkers in dogs. The study finds that supplementing dog diets with olive flour rich in polyphenols can significantly decrease oxidative stress, as measured by d-ROMs, and shows an increasing trend in blood antioxidants. This research contributes to understanding the potential benefits of using olive oil by-products in pet nutrition and highlights the positive implications for animal health and environmental sustainability.

Line 43: The term "phenolic rings" should be further explained, as it may not be immediately clear to all readers what these compounds are and their relevance to oxidative stress.

AU: correction made—(now l 43-48)

Line 85: The inclusion criteria for the dogs (weight and age range) should be justified in the context of the study. It would be helpful to explain why these specific parameters were chosen.

 AU: correction made —(now l91-93)

“The nutritional trial was carried out at two shelter kennels located in the district of Pisa and Lucca (central Italy), both of which maintained identical health, feeding, and space management. “ the similar management reduce can reduce variability factor. “ The dogs were recruited were similar for weight (between 15 and 30 kg) and aged between 2 and 8 years (not in the growing phase nor aging)

How reported the nutritional trial was carried out at two shelter kennels located in the district of Pisa and Lucca (central Italy), both of which maintained identical health, feeding, and space management in order to avoid . Thirty dogs were recruited following clinical evaluation, similar for weight (between 15 and 30 kg) and aged between 2 and 8 years (not in the growing phase nor aging).

Line 102: To enhance clarity and provide context for all readers, it would be beneficial to include the full meaning of the acronym AOAC in the manuscript.

 AU: correction made —(now  l 109)

Line 145: Specify the exact statistical methods used in the ANOVA analysis. Clarifying whether assumptions were checked (e.g., normality, homoscedasticity) would enhance the scientific rigor.

 AU: correction made —(now l. 152-153)

Line 155 (Table 2): it would be helpful to include a footer that provides the full meanings of all acronyms used. This addition would enhance the clarity of the table and ensure that all readers, regardless of their familiarity with the specific terminology, can fully understand the data presented.

 AU: correction made —(now l. 165-168)

Line 178: The reported 16% reduction in d-ROMs should include confidence intervals to provide a sense of the variability and precision of this estimate.

 AU: correction made —(now l. 165-168)

Abstract

The abstract effectively summarizes the study's aim, methodology, key findings, and implications. It concisely presents the novel aspect of the research, using olive pulp flour in dog diets, and highlights the significant reduction in oxidative stress markers. However, it could benefit from more detailed information on the study's methodology and the broader impact of the findings on pet nutrition and sustainability.

 AU: the abstract should be a maximum of 200 words, we have however added a sentence at l 31-33

We refer to the publisher to decide whether to add it

Introduction

The introduction provides a clear background on the importance of utilizing olive oil by-products and their potential health benefits. It effectively sets the context for the study and highlights the novelty of applying these by-products to dog diets.

AU: The authors thank the referee for the comment

Materials and methods

The statistical analyses used in the manuscript are generally appropriate for the study design and objectives. However, to enhance the rigor and transparency of the research, it would be advisable to include more details on the assumptions of the statistical tests and supplementary metrics such as confidence intervals. These additions would provide a more comprehensive understanding of the data and its implications.

 AU: thank you, more details have been added (line 155-156); confidence intervals were already presented

Results

The results section clearly presents the findings, with tables providing detailed data on oxidative stress markers. Providing confidence intervals or effect sizes alongside p-values would strengthen the interpretation of the results.

AU: Mean values and confidence intervals were already presented in the table 2

Discussion

The discussion appropriately interprets the results in the context of existing literature, highlighting the significance of the findings and their implications for pet nutrition and sustainability. The authors make relevant connections to the potential benefits of olive flour for reducing oxidative stress in dogs. However, the discussion could be improved by addressing the limitations of the study, such as the sample size and the generalizability of the findings, and by suggesting directions for future research.

 AU: The authors have highlighted the limitation of the study and future research at lines 191-192

Conclusion

While the conclusion mentions the potential for economic and environmental benefits, the specifics of how olive pulp flour contributes to these areas could be expanded. For example, quantifying potential waste reduction or cost savings would provide concrete evidence of these benefits.

AU: Currently no specific techno-economic analyses and life cycle assessments are available about of the use of Olive pulp flour in pet feed. Therefore, some lines of the conclusions have been changed (l 233-243)

Reviewer 2 Report

Comments and Suggestions for Authors

In the present study, the authors investigated the influence of olive meal (MOP®) on oxidative blood biomarkers in dogs. Thirty dogs were recruited and divided into two groups. Both groups received the same food twice daily in the form of kibble and canned diet. The treatment group (T) received an addition of 11.5 mg/kg organic olive meal daily. The results showed that d-ROMs decreased significantly in the treatment group. There are a few things to improve:

1. The discussion is too short and should be more detailed – to compare papers that supplement other nutrients with similar antioxidant effects and their results.

2. There is a lack of analysis of supplementation – both a Weende analysis and an analysis of key antioxidants should be provided.

3. The diets of the control and treatment dogs should be compared. Were they fed an isoenergetic diet?

Author Response

Answers to the Reviewer 2

Comments and Suggestions for Authors

In the present study, the authors investigated the influence of olive meal (MOP®) on oxidative blood biomarkers in dogs. Thirty dogs were recruited and divided into two groups. Both groups received the same food twice daily in the form of kibble and canned diet. The treatment group (T) received an addition of 11.5 mg/kg organic olive meal daily. The results showed that d-ROMs decreased significantly in the treatment group. There are a few things to improve:

  1. The discussion is too short and should be more detailed – to compare papers that supplement other nutrients with similar antioxidant effects and their results.

AU: the authors would be glad to modify the discussion if the reviewer could give more precise suggestions or provided literature references. The comment seems too generic

  1. There is a lack of analysis of supplementation – both a Weende analysis and an analysis of key antioxidants should be provided.

AU: composition of supplementation was previously analysed and reported by Cecchi et al. [21].

  1. The diets of the control and treatment dogs should be compared. Were they fed an isoenergetic diet?

AU: as already reported “Both groups were fed the same dry commercial pet food and wet food twice daily ac-cording to individual maintenance energy requirements, with the only difference being the supplementation of organic olive flour (MOP-ManniOlivePowder®) rich in poly-phenols”. Therefore the

Dogs fed isoenergetic diets.

Reviewer 3 Report

Comments and Suggestions for Authors

2.2 Diet composition --> I didn't understand the diet composition in the two groups: do they use the same commercial dry and wet diets in the two shelters for all the dogs in this study? Perhaps this should be made clearer.

Conclusions: In my opinion, the conclusions should be checked. Only 1 out of 3 of your outcome measures shows significant improvement, and as you say in the discussion section, this could be partly related to the time of administration, but we are not really sure. Maybe a good improvement for the reader, as already suggested by different authors, is really the question of time: we cannot expect a change of mayor in a short period of time, and that is fine. 

I like the circular economy part and that is fine, but the results should be better reported in the conclusions.

Author Response

The authors would like to thank the Editor and the Reviewers for their time and comments that helped improve the manuscript.

Answers to the Reviewer 3

Comments and Suggestions for Authors

2.2 Diet composition --> I didn't understand the diet composition in the two groups: do they use the same commercial dry and wet diets in the two shelters for all the dogs in this study? Perhaps this should be made clearer.

AU: Yes the two groups used the same commercial dry and wet diets, this sentence was clarified

Conclusions: In my opinion, the conclusions should be checked. Only 1 out of 3 of your outcome measures shows significant improvement, and as you say in the discussion section, this could be partly related to the time of administration, but we are not really sure. Maybe a good improvement for the reader, as already suggested by different authors, is really the question of time: we cannot expect a change of mayor in a short period of time, and that is fine.

AU: Thank you for the comment. A sentence has been added in the conclusions (l 230)

I like the circular economy part and that is fine, but the results should be better reported in the conclusions.

AU: some lines of the conclusions have been changed (233-243). Since no specific techno-economic analyses and life cycle assessments are available about of the use of Olive pulp flour in pet feed the author preferred to change this part

Round 2

Reviewer 1 Report

Comments and Suggestions for Authors

Thank you for the revisions you have made in response to the feedback provided. I appreciate your careful attention to the suggestions, and it is clear that the manuscript has significantly improved in clarity and scientific rigor.

Thank you for your hard work and responsiveness. I look forward to seeing this research published and contributing to ongoing discussions in our field.